# Factory Crystallized Silicates for Monolithic Metal-Free Restorations: A Flexural Strength and Translucency Comparison Test

**DOI:** 10.3390/ma15217834

**Published:** 2022-11-06

**Authors:** Alessandro Vichi, Zejiao Zhao, Gaetano Paolone, Nicola Scotti, Mahdi Mutahar, Cecilia Goracci, Chris Louca

**Affiliations:** 1Dental Academy, University of Portsmouth, William Beatty Building, Hampshire Terrace, Portsmouth PO1 2QG, UK; 2Department of Dentistry, IRCCS San Raffaele Hospital and Dental School, Vita Salute University, 20132 Milan, Italy; 3Department of Surgical Sciences, Dental School Lingotto, University of Turin, 10126 Turin, Italy; 4Department of Medical Biotechnologies, University of Siena, 53100 Siena, Italy

**Keywords:** ceramic, metal-free, CAD/CAM, flexural strength, translucency, lithium silicate, lithium disilicate

## Abstract

Flexural strength (FS) and translucency (Contrast Ratio-CR) of three different factory crystallized silica-based glass ceramics, Celtra Duo (CD), N!ce (NI) and Li-Si Block, a lithium disilicate, IPS e.max CAD (LD), and a leucite-reinforced feldspathic ceramic, Empress CAD (EM), in two different translucencies (HT and LT) for use in chairside dental restorations have been compared. CAD blocks of the materials were cut into beams and tiles and processed following manufacturers’ instructions. The beams were tested (3-PBT) to determine flexural strength, Weibull characteristic strength, and Weibull modulus; and tiles were tested to determine CR. All data were statistically analyzed. In addition, SEM analysis of the materials was performed. Differences in flexural strength (FS) and translucency (CR) between the materials were found to be statistically significant. FS decreased as follows (MPa): LDHT 350.88 ± 19.77 (a) = LDLT 343.57 ± 18.48 (a) > LSLT 202.15 ± 17.41 (b) = LSHT 196.93 ± 8.87 > NIHT 186.69 ± 13.06 (c) = CDLT 184.73 ± 13.63 (c) = CDHT 174.15 ± 21.76 (c) = NILT 172.12 ± 11.98 (c) > EMHT 131.16 ± 13.33 (e) = EMLT 127.65 ± 11.09. CR decreased as follows (mean ± sd): CDLT 74.1 ± 1.1 (a); LSLT 74.0 ± 1.1 (ab); NILT 73.3 ± 0.8 (ab); EMLT 73.0 ± 1.5 (ab); NIHT 72.4 ± 1.0 (bc); LDLT 71.3 ± 1.1 (bc); LTHT 65.2 ± 0.9 (de); LSHT 63.8 ± 1.1 (def); EMHT 636 ± 1.2 (ef); CDHT 62.2 ± 0.8 (f). Our findings show that factory-crystallized lithium silicate glass ceramics fulfill ISO standards for Classes 1 and 2. Therefore, they can be considered viable alternatives to produce single-unit restorations with a chairside procedure not requiring thermal treatment.

## 1. Introduction

Metal-free dental restorations are continuously expanding in comparison to porcelain fused to metal restorations thanks to their improved accuracy, easy-of-use, and digital workflow integration. Materials have also been improved in strength and aesthetics. Among the available materials, lithium silicates are outstanding in the CAD/CAM scenario. Good mechanical properties together with satisfactory aesthetic performance have allowed this material to gain in popularity over time.

The Cerec system was initially developed with the aim of completing a dental ceramic restoration in a single appointment [1,2,3], with the entire manufacturing process carried out within the dental office. Ongoing improvements in this system have led to a wider acceptance in dental practice [4,5,6]. Alongside hardware and software improvements, materials have been improved and/or newly developed. For the Cerec system, several materials are available, and the selection criteria are related to clinical use [7], with mechanical and optical properties as pivotal. Among these materials, some require additional manufacturing processes like sintering that need to be performed in dedicated furnaces usually not present in dental offices. Other materials such as traditional lithium disilicate also require an additional time-consuming manufacturing process (crystallization) that needs to be performed in a furnace, even if not specifically dedicated, which makes them far from being defined as chairside materials. Conversely, feldspathic and leucite-reinforced ceramics require only finishing and polishing procedures that can be performed manually, so they are indicated for chairside use. Feldspathic materials have a number of advantages despite having lower mechanical properties than polycrystalline and silicate glass ceramics. These advantages include the positive characteristics of all-ceramic restorations, such as aesthetic appearance, biocompatibility, and durability [8], with the benefits of being manufactured by CAD/CAM, such as time savings, cost-effectiveness, and quality control [9,10,11].

More recently, new silicate glass ceramics have been developed. They are crystallized by the manufacturers (Factory Crystallized, FC), and they do not require thermal treatment in a furnace. They have been produced with the aim of combining the ease-of-use of feldspathic and leucite-reinforced feldspathic ceramics with the resistance of the silicates requiring a furnace-based thermal treatment. These new materials are claimed to be likely to progressively replace feldspathic and leucite-reinforced glass ceramics because they have a similar finishing process, non-requiring thermal treatment, but improved resistance. Strength and translucency are the most important properties that influence the clinicians’ selection of materials. Particularly, flexural strength can be considered a relevant mechanical property for brittle materials that are much weaker in tension than in compression [12,13], and it is measured in several ways, with the three-point bending test being the most popular. Translucency is considered one of the most important factors in matching the appearance of natural teeth with restorative materials and is defined as the relative amount of light transmission. At clinically indicated thicknesses, these materials do not offer a complete barrier to light transmission through the structure, unlike porcelain fused to metal restorations [14]. The presence of a glass matrix results in higher translucency when compared with, for example, sintered polycrystalline zirconia or other ceramic materials [15]. The aim of this study was to measure some of the mechanical properties of these factory-crystallized lithium silicate glass ceramics including flexural strength (*σ*), Weibull modulus (*m*), and Weibull characteristic strength (*σ*_0_). In addition, as CAD/CAM blocks are generally available in different translucencies, usually low translucency (LT) or high translucency (HT), to accomplish different clinical situations, the optical properties in terms of translucency, measured by the contrast ratio, were assessed. Three different factory-crystallized lithium silicates were compared to a lithium disilicate glass ceramic requiring thermal treatment, considered as the alternative ceramic material for monolithic single tooth restorations [16], and with a leucite-reinforced glass ceramic not requiring thermal treatment. The tested null hypotheses were (i) there were no statistically significant differences in terms of flexural strength between the tested materials, and (ii) translucency between the tested materials is different between LT and HT formulations but not between the tested materials.

## 2. Materials and Methods

Three CAD/CAM “factory crystallized” (FC) materials in two different translucencies each (high translucency and low translucency) were selected for the study: Celtra Duo (Denstsply Sirona, Charlotte, NC, USA); n!ce (Straumann, Basel, Switzerland), and Initial LiSi Block (GC, Tokyo, Japan). Lithium disilicate IPS e.max CAD (Ivoclar Vivadent AG, Schaan, Liechtenstein) in HT and LT translucencies was used as a control material for lithium silicates requiring thermal treatment. IPS Empress CAD (Ivoclar Vivadent AG, Schaan, Liechtenstein) in HT and LT translucencies was used as a control material for chairside materials not requiring thermal treatment. The blocks of all materials used in this study were in C14 size and A3 Vita shade. With the use of a proprietary device, blocks were perpendicularly cut to obtain the desired shape depending on the test to be performed, flexural strength, or translucency. The e.max CAD specimens were submitted to crystallization firing in a ceramic furnace (EP 600 Combi, Ivoclar Vivadent AG, Schaan, Liechtenstein) following the manufacturer’s instructions. The composition of the materials selected is given in Table 1.

### 2.1. Flexural Strength

For the three-point bending test, beam-shaped specimens (n = 15 per group) were cut with a water-cooled cutting machine (Low-speed saw, Buehler, Lake Bluff, IL, USA), and wet-finished in a grinder/polisher machine (Minimet, Buehler, Lake Bluff, IL, USA) with 600 grit paper until dimensions of 1.2 ± 0.2 mm thickness by 4.0 ± 0.2 mm width, and 15.0 ± 0.2 mm were obtained. Specimens were subsequently wet-polished with 1200 and 2400 grit paper. According to ISO 6872:2015, a 45° edge chamfer was made at each major edge (ISO 6872:2015) [17]. Specimens were ultrasonically cleaned in distilled water for 10 min before the measurement procedure. Tests were performed in a universal testing machine (ESM 303, Mark-10, Copiague, NY, USA) equipped with a 50 N load cell (M5-50, Mark-10, Copiague, NY, USA) with a crosshead speed of 1 mm/min. The span was set at 13.0 mm. Specimens were tested dry at room temperature. The fracture load was recorded in N, and the flexural strength (*σ*) was calculated in MPa by using the following equation [9,10]:$$\sigma =\frac{3Pl}{2wb^{2}}$$
where: *P* is the fracture load in N, *l* is the distance between the center of the supports in mm, *w* is the width in mm, and *b* is the height in mm (Figure 1).

The Weibull characteristic strength (*σ*_0_) and the Weibull modulus (*m*) were calculated according to the following equation [9,10]:$$P_{f}=1-\exp\left[-{\left(\frac{\sigma }{\sigma_{0}}\right)}^{m}\right]$$
where: *P_f_* is the probability of failure between 0 and 1, *σ* is the flexural strength in MPa, *σ*_0_ is the Weibull characteristic strength in MPa, and *m* is the Weibull modulus.

### 2.2. Translucency Measurements

For optical evaluation, tab-shaped specimens (n = 10 per group) were cut with a water-cooled cutting machine (Low-speed saw, Buehler, Lake Bluff, IL, USA), and wet-finished in a grinder/polisher machine (Minimet, Buehler, Lake Bluff, IL, USA) with 600 grit and 1200 grit paper in a grinder/polisher machine (Minimet, Buehler, Lake Bluff, IL, USA) until final dimension of 15.0 ± 0.5 mm in length, 15.0 ± 0.5 mm in width, and 1.0 ± 0.1 mm in thickness were obtained. Specimens were ultrasonically cleaned in distilled water for 10 min before the measurement procedure. The measurements were performed using a colorimeter (Color Meter Pro, Vetus Technology Co., Hefei, China) with an 8 mm aperture and D/8 geometry of viewing. The colorimeter was connected to a smartphone running a dedicated app (ColorMeter V2.1.27). D65 illumination and 10 degrees standard observation angle were selected. Data were recorded in the CIEXYZ colorimetric system. A quantitative measurement of translucency was made by comparing the reflectance of light “Y” in the CIEXYZ colorimetric system (ratio of the intensity of reflected radiant flux to that of the incident radiant flux) through the test specimen over a backing with high reflectance (white backing = Yw) to that of low reflectance or high absorbance (black backing = Yb). For every specimen, the Contrast Ratio was calculated with the following equation [18]:


CR = Yb/Yw.

### 2.3. SEM Analysis

An extra specimen per group was produced for microscopic glass ceramic microstructural evaluation. Specimens were etched for 120 s with 9% hydrofluoric acid (Porcelain Etch, Ultradent, South Jordan, UT, USA), and cleaned under running water. The preparation for SEM analysis involved ultrasonically cleansing in a 95% alcohol solution for 3 min and air drying with an oil-free air spray. Specimens were then secured onto SEM (Tescan MIRA 3 FEG-SEM, Brno, Czechia) slabs with gold conducting tape, and gold coated in a vacuum sputter coater (Quorum Q150R sputter coater, Quorum Technologies, Laughton, UK). The treated surfaces were then observed at 20,000× magnification for morphology evaluation.

### 2.4. Statistical Analysis

Flexural Strength

Data were tested to fit a normal distribution with the Kolmogorov–Smirnov test and the homogeneity of variances was verified with Levene’s test. According to these preliminary tests, a one-way ANOVA was performed, followed by the Tukey test post hoc.

Translucency (CR)

Data were tested to fit a normal distribution with the Kolmogorov–Smirnov test and the homogeneity of variances was verified with Levene’s test. According to these preliminary tests, a one-way ANOVA was performed, followed by the Tukey test post hoc.

In all the statistical tests, the level of significance was set at *p* < 0.05. The statistical analyses were processed by SigmaPlot 11.0 (Systat Software, Inc., San Jose, CA, USA) software.

## 3. Results

### 3.1. Flexural Strength

The mean of the flexural strength(*s*), Weibull characteristic strength (*σ*_0_), Weibull modulus (*m*), and the statistical significance values are reported in Table 2. The one-way ANOVA revealed a statistically significant difference (*p* < 0.001).

e.max CAD HT with 350.88 ± 19.77 MPa and e.max CAD LT with 343.57 ± 18.48 MPa reached the highest levels of flexural strength and the difference with the other materials was statistically significant. This material in both opacities was the only one of those tested to fulfill Class 3 ISO 6872:2015 indications, specifically monolithic single-unit restorations non-adhesively cemented and partially or fully covered substructures not involving posterior restoration, limited to three-unit prostheses. Initial LiSi Block in both LT and HT showed statistically significant differences with n!ce in both LT and HT and Celtra Duo in both LT and HT. The latter four groups were not statistically significantly different from each other. IPS Empress CAD HT with 131.16 ± 13.33 and IPS Empress CAD LT with 127.65 ± 11.09 showed the lowest flexural strength, statistically significantly different from the other eight groups. However, this material in both opacities fulfilled the indications for Classes 1 and 2 of ISO standard 6872:2015 [17].

All the tested materials achieved a flexural strength higher than 100 MPa, required for classes 1 and 2 of ISO standard 6872:2015 [17], thus for the clinical indications of single-tooth monolithic ceramic or substructure adhesively cemented (Table 3).

### 3.2. Translucency

CR values and statistical significance are reported in Table 4. The one-way ANOVA revealed a statistically significant difference (*p* < 0.001). The materials were in the following order from most opaque to most translucent (mean ± standard deviation). Different letters in parentheses label statistically significant differences: Celtra LT 74.1 ± 1.1 (a); LiSi LT 74.0 ± 1.1 (ab); n!ce LT 73.3 ± 0.8 (ab); Empress CAD LT 73.0 ± 15 (ab); n!ce HT 72.4 ± 1.0 (bc); e.max CAD LT 71.3 ± 1.1 (bc); e.max CAD HT 65.2 ± 0.9 (de); LiSi HT 63.8 ± 1.1 (def); Empress CAD HT 63.6 ± 1.2 (ef); Celtra HT 62.2 ± 0.8 (f). ΔT between HT and LT formulations was 9.4 for Empress CAD, 6.1 for e.max CAD, 11.9 for Celtra, 10.2 for LiSi Block, and 0.9 for n!ce.

### 3.3. SEM Analysis

By conditioning with HF gel, the glassy phase superficially dissolved, and the crystalline phase eventually became visible. Empress CAD showed the typical porous structure of a feldspathic ceramic, with large voids. e.max CAD showed the typical club-like crystalline microstructure of lithium disilicate glass-ceramic. Celtra Duo showed shorter, densely distributed, round-shaped crystals, typical of zirconia-reinforced lithium silicates. n!ce showed an original structure, different from that observed before in other lithium (di)silicate materials, in which short needle-like crystals are present in a small amount surrounded by globular porous structures. LiSi Block showed an original structure as well, with a prevalent homogeneous presence of a large amount of short needle-shape crystals and some sprout-shape formations, which were slightly porous and regularly distributed. These sprout-shaped formations are likely the results of the aggregation of lithium disilicate crystals and glass matrix after etching. Due to the smaller crystal size present in LiSi blocks, compared to lithium disilicate, combined with a higher acid resistance of the glass matrix, these aggregations are more present in LiSi Blocks. For Empress CAD, LiSi and n!ce the differences between HT and LT formulations were not noticeable. For e.max CAD, HT formulation showed longer and more regularly distributed crystals. For Celtra, the HT formulation showed smaller crystals than the LT (Figure 2).

## 4. Discussion

The flexural strength test results showed a statistically significant difference between the groups tested, therefore the first null hypothesis has to be rejected. Translucency was supposed to be the only difference between the HT and LT formulations, but the results showed statistically significant differences between materials and one material (n!ce) in which the difference was not statistically significant between the LT and HT formulations; therefore the second null hypothesis has also to be rejected.

Flexural strength can be considered a relevant mechanical property for brittle materials that are much weaker in tension than in compression. Common ways to assess this property are the three-point bending test (3-PBT), the four-point bending test (4PBT), and the biaxial flexure test (BFT) (ISO 6872:2015) [17]. For this study, the 3-PBT was selected as it is the most commonly performed test due to its higher standardization and easier setup thus allowing a higher possibility of comparison with other studies. However, 3-PBT is a monotonic test in which the load is applied until specimen failure. This is a limitation of the study as it is not representative of the clinical situation in which the restoration is subjected to cycle load and thermal variations. In the present study, e.max CAD showed the highest flexural strength, statistically different from the other four groups. The values obtained, respectively 350.88 ± 19.77 MPa for HT translucency and 343.57 ± 18.48 MPa for LT, are in line with other findings such as 336.06 ± 40 MPa for HT and 376.85 ± 39.09 MPa for LT reported by Sedda et al. [10], 367 ± 44 MPa by Lien et al. [19], 348.33 MPa by Elsaka et al. [20], 377 MPa by Carrabba et al. [21], 356.7 ± 59.6 MPa by Homaei et al. [22], 378.88 ± 55.3 MPa by Juntavee et al. [23], 341.88 ± 40.25 MPa by Leung et al. [24], and 362 ± 78.6 MPa by Lien et al. [19]. It should be noted that in the present study the standard deviation was lower than in other studies. This is likely due to the methods and devices used for the test, which can greatly affect the outcome [25]. In Table 3 the recommended clinical indications from ISO standard 6872:2015 [17] are reported. Based on these recommendations, e.max CAD fulfilled the requirements for Class 3, that is for three-unit prostheses not involving molar restorations, both monolithic as well as substructure. e.max CAD was the only material that was able to reach the values for this Class 3 classification. LiSi Block, a factory-crystallized lithium silicate that does not require thermal treatment after milling, obtained in the present study 202.15 ± 17.41 MPa for LT translucency and 196.93 ± 8.87 MPa for HT. In the literature, at present no data are available for this material tested with 3-PBT, 4-PBT or biaxial tests. The differences with the other two silicates not requiring thermal treatment were statistically significant; n!ce, a lithium aluminosilicate, (186.69 ± 13.06 MPa for HT and 172.12 ± 11.98 MPa for LT) and Celtra, a zirconia reinforced lithium silicate (184.73 ± 13.63 MPa for HT and 174.15 ± 21.76 MPa for LT) obtained lower values than LiSi Block. The differences between n!ce and Celtra Duo in the two different translucencies were not statistically significant. Concerning n!ce, only one study was found in the literature testing this material for flexural strength, but it is not directly comparable to the present study as it has been performed with the biaxial flexural strength test [26]. However, as the biaxial is considered to give value higher by about 15–20% than 3-PBT [27], the reported values of 222 ± 28.4 MPa for LT seem in line with the data of the present study. Concerning Celtra, Kim et al. [28] reported 258.92 ± 31.17 MPa, while Lawson et al. [29] reported 300.1 ± 16.8 MPa. These results look higher than those obtained in the present study. However, in both of the cited studies, the dimensions of the specimens were not in accordance with ISO 6872:2015 standards. Conversely, the data reported by Riquieri et al. [30] (163.86 MPa) and Stawarczyk et al. [26] (190 ± 23.3 MPa for LT and 177 ± 29.1 MPa for HT), even if tested with the biaxial test, seem in general accordance with the data of the present study. Empress CAD, a leucite-reinforced glass ceramic, obtained for flexural strength the lower values among the materials tested (131.16 ± 13.33 MPa for Empress CAD HT and 127.65 ± 11.09 MPa for LT), and the difference with the other materials was statistically significant. Empress CAD was selected as the control group in the present study as in a previous study on chairside materials [9] it obtained the highest results among the feldspathic ceramics (125.10 ± 13.05 MPa). As feldspathic materials do not require thermal treatment, this material could be considered representative of feldspathic materials not requiring thermal treatment, thus allowing an effective chairside procedure. It should be noted that notwithstanding the statistically significant differences, all four materials not requiring thermal treatment (LiSi Block, n!ce, Celtra Duo, and Empress CAD) fulfill the requirements for Class 2 of ISO standards 6872:2015, that is monolithic ceramic for a single unit, anterior or posterior prostheses, adhesively cemented. At the same time, none of the four satisfied the requirements for Class 3, which were obtained only by e.max CAD.

The differences between the materials tested could be ascribed mainly to the crystal phase. For IPS e.max CAD, on heating, the metasilicate phase dissolves and lithium disilicate crystallizes. This reaction is controlled by the nucleating agents. As can be seen in the SEM analysis (Figure 2c,d), this reaction results in the presence of a high number of elongated crystals. For the three factory-crystallized materials, the crystals are present but in a smaller number and size. This morphology does not allow the crystals to form an interlocked microstructure, thus it is likely that the mechanical properties of the ceramic decrease.

Concerning the Weibull modulus and the Weibull characteristic strength, they serve as a more accurate representation of the structural reliability of dental ceramics. The Weibull characteristic strength (*σ*_0_), indicates the 63.21 percentile of the strength distribution, while the Weibull modulus (*m*), is an indication of the distribution of flaws. Generally speaking, it is often preferable to have a higher *m*, even with a slightly lower mean fracture strength, than a lower *m* associated with a higher mean fracture strength. Materials with higher Weibull moduli have a more uniform distribution of defects which describes the lifetime and frequency of failure of brittle materials. In particular, an *m* greater than 20 indicates a higher level of the structural integrity of the material and a higher reliability. In the present study only e.max CAD HT (20.95), e.max CAD LT (22.09), and LiSi Block HT (26.88) had *m* values greater than 20. The data collected in the present study are generally in line with the *m* values of brittle materials, which have been reported to be in the 5–15 range.

Translucency is one of the main parameters in matching the appearance of the restoration and the natural tooth and was identified as a pivotal factor in controlling aesthetics and is a critical consideration for material selection [18].

The translucency of a material directly involves three parameters: the contrast ratio (CR), transmittance, and translucency parameter (TP). CR has been selected in the present study as its use is broad in dental literature, making it easier to compare results. CR is the ratio of the reflectance of a specimen over a black backing to that over a white backing of known reflectance and is an estimate of opacity. CR ranges from 0 to 100 (or from 0 to 1, or from 0% to 100%), with 0 corresponding to transparency (totally translucent) and 1 corresponding to total opacity (absence of translucency). The mean measured values of CR of e.max CAD, the only fully crystallized material, were similar to those reported in the literature [9]. Conversely to other materials whose findings are reported in the literature [9,31,32], the difference in CR between the various LT translucencies was consistent, with the most opaque among the LT being Celtra Duo with 74.1 ± 1.1 and the less opaque e.max CAD 71.3 ± 1.1. This similarity is a convenient characteristic for clinicians as it allows them to switch between materials with reduced problems from an optical viewpoint as they behave similarly. The HT translucencies showed a different path. Four out of five materials showed again consistency, e.max CAD HT, LiSi Block HT, Empress CAD HT, and Celtra HT in a range between 65.2 ± 0.9 and 62.2 ± 0.8. n!ce was an exception as the HT translucency showed a CR of 72.4 ± 1.0, very close to that of the LT translucency (73.3 ± 0.8). This finding has to be taken into consideration in the clinical selection of materials as the ΔT between LT and HT translucencies of the other 4 materials was 6.1 for e.max CAD, 9.4 for Empress CAD, 10.2 for LiSi Block, and 11,9 for Celtra, while concerning n!ce it was only 0.9, far below the threshold for clinical perceptibility. Differences below 7 in CR should be in fact considered not visible by the human eye based on the mean translucency perception threshold (TPT) defined by Liu et al. [33]; although the authors recognized that there were significant variations depending upon the observer, e.g., a clinician with 10 years of shade-matching experience could have a TPT of 4. For this reason, the two translucencies of n!ce can be considered from an optical viewpoint two-of-a-kind. It has to be underlined that the reported CR for Enamel and Dentine is about 45 and 65 respectively [34], therefore all the materials showed a translucency closer to the one of the dentin than to the enamel, even if the variation of the translucency due to thickness has to be taken into account. A limitation of this study, in common with the vast majority of the studies, is that only one thickness has been tested for CR and a more extensive investigation comprising more thicknesses can be more representative of the optical behavior of the materials. Identifying the reasons for the differences in the translucency of the different materials is not straightforward, and the SEM analysis performed cannot be considered completely explicative in this regard. Even if it is known that by adding nucleating agents the shape of the crystals can be controlled, and that at a higher density of crystals the translucency decreases, the differences in translucency among the various materials are probably the result of the complex interaction between size, shape, and number of crystals, and glass phase.

The well-known relationship that correlates mechanical properties, translucency, and material thickness should be carefully evaluated by clinicians during material selection. Lowering the thickness of the restoration would allow the material to be more translucent [35,36]. At the same time, the minimal indicated thickness should always be respected in order to avoid the risk of fracture. Precise indications for minimal thickness should be provided by the manufacturer with respect to the wide range of available materials in fixed prosthodontics.

The present test evaluated several materials for their strength and translucencies. However, as a further limitation of the study, these investigations cannot be considered completely representative of the clinical behavior, especially due to the absence of cementation. Adhesive cementation in fact enhances the possibility of reducing the ceramic thickness in single restorations, but this aspect is under-investigated.

As clinical indications, it can be concluded that the three silicates not requiring thermal treatment tested cannot be considered at present a replacement of the lithium disilicate requiring thermal treatment, which still retains higher strength values and allow the fabrication of three-unit prostheses. However, they can be considered a valid alternative to lithium disilicate for single crowns. Particularly, as they do not require thermal treatment, they can be considered a chairside material thus a replacement of feldspathic glass ceramics, allowing a similar working time with increased mechanical performance.

## 5. Conclusions

Within the limitation of this in vitro study, the following conclusions could be drawn:

The flexural strengths of the materials tested were statistically different. The lithium disilicate that requires thermal treatment (e.max CAD) achieved statistically significantly higher results than the factory crystallized lithium silicates (LiSi Block, n!ce, Celtra). The three FC lithium silicates achieved significantly higher flexural strength than leucite-reinforced glass ceramic Empress CAD.

Among the three factory-crystallized lithium silicates, LiSi Block achieved statistically significant higher flexural strength values.

Considering the ISO standard categories of ISO 6872:2015, e.max CAD, the only material requiring thermal treatment, reached values higher than 300 MPa and can be therefore indicated for up to three-unit FPDs. All the other materials were in the range of 100 MPA to 300 MPa, with the recommended clinical indication for monolithic single-unit prostheses adhesively cemented.

Concerning translucency, all the materials showed a difference of at least ΔT = 6.1 between HT and LT formulation, over the TPT for experienced operators (ΔT = 4), except n!ce for which the difference was ΔT = 1, below the limits of perception.

## Figures and Tables

**Figure 1 materials-15-07834-f001:**
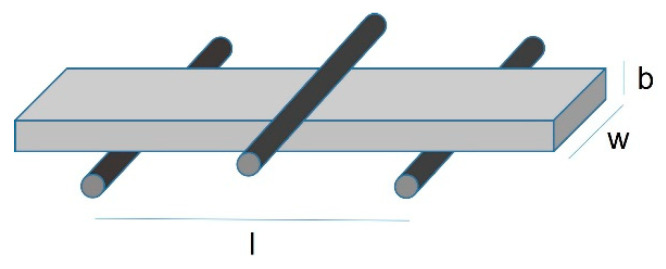
Schematic representation of the 3-PBT flexural strength testing setup.

**Figure 2 materials-15-07834-f002:**
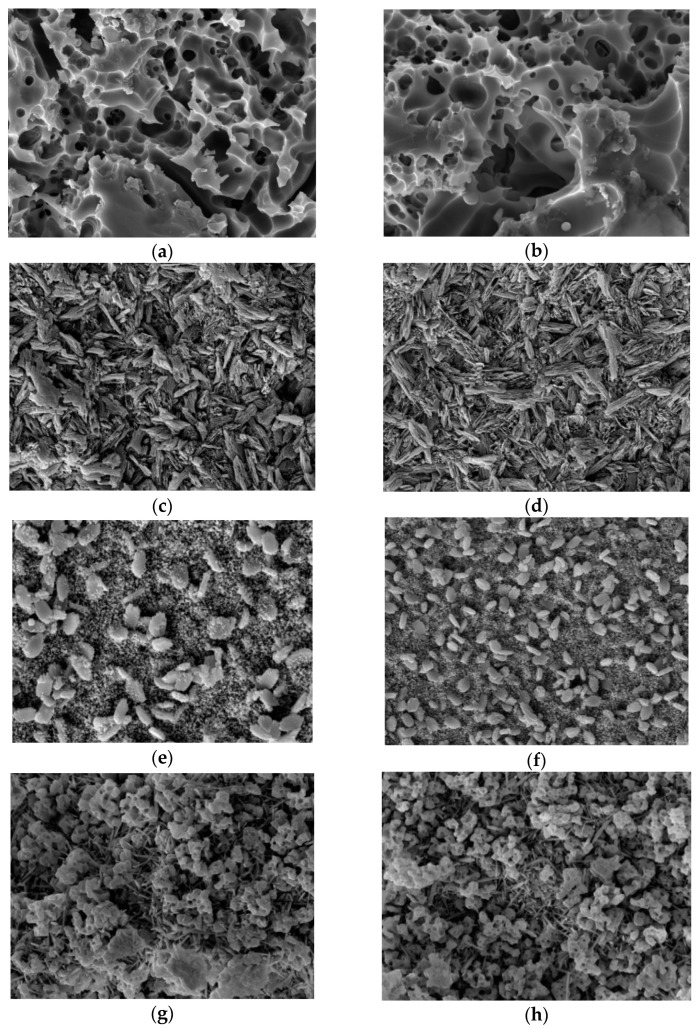
SEM pictures of the materials tested in both LT and HT formulations (20.000×). (**a**) Empress CAD LT; (**b**) Empress CAD HT; (**c**) e.max CAD LT; (**d**) e.max CAD HT; (**e**) Celtra Duo LT; (**f**) Celtra Duo HT; (**g**) n!ce LT; (**h**) n!ce HT; (**i**) LiSi Block LT; and (**j**) LiSI Block HT.

**Table 1 materials-15-07834-t001:** Composition of the materials tested (manufacturers’ data).

Materials	Code	Manufacturer	Translucency	Chemical Composition	Manufacturer Definition	Thermal Treatment
Celtra Duo	CD	Dentsply Sirona	HT/A3/C14; LT/A3/C14	58% SiO_2_; 18.5% Li_2_O; 5% P_2_O_5_; 10.1% ZrO; 1.9% Al_2_O_3_; 2% CeO_2_; 1% Tb_4_O_7_	lithium silicate Zirconia reinforced.	Optional *
Initial LiSi	LS	GC Europe	HT/A3/C14; LT/A3/C14	55–80% SiO_2_, 10–30% Li_2_O; 5–20% other oxides; pigments: trace **	lithium disilicate	NO
n!ce	NI	Straumann	HT/A3/C14; LT/A3/C14	64–70% SiO_2_; 10.5–12.5% Li_2_O; 0–3% K_2_O; 3–8% P_2_O_5_; 0–0.5% ZrO_2_; 10.5–11.5% Al_2_O_3_; 1–2% CaO; 0–9% pigments;1–3% Na_2_O	lithium aluminosilicate ceramic reinforced with lithium disilicate	NO
IPS e.max CAD	LD	Ivoclar Vivadent	HT/A3/C14; LT/A3/C14	57–80% SiO_2_; 11–19% Li_2_O; 0–13% K_2_O; 0–11% P_2_O_5_; 0–8% ZrO_2_, 0–8% ZnO; 0–12% others + coloring oxides	lithium disilicate	YES
IPS Empress CAD	EM	Ivoclar Vivadent	HT/A3/C14; LT/A3/C14	60–65% SiO_2_; 16.0–20.0% Al_2_O_3_; K_2_O 10.0–14.0%; Na_2_O 3.5–6.5%, Other oxides 0.5–7.0%, pigments 0.2–1.0%	leucite-reinforced glass ceramic	NO

* In the present study thermal treatment was not performed. ** Company personal communication.

**Table 2 materials-15-07834-t002:** Results of tested materials ordered by flexural strength.

		Flexural Strength		Weibull Statistics	
Material	Type	*σ* (MPa)	*Sig*	*m*	*σ*_0_ (MPa)
e.max CAD HT	lithium disilicate	350.88 ± 19.77	a	20.95	359.82
e.max CAD LT	lithium disilicate	343.57 ± 18.48	a	22.09	351.85
Initial LiSi Block LT	lithium disilicate	202.15 ± 17.41	b	13.52	209.91
Initial LiSi Block HT	lithium disilicate	196.93 ± 8.87	b	26.88	200.91
n!ce HT	lithium Aluminosilicate	186.69 ± 13.06	c	16.82	192.59
Celtra Duo LT	lithium Silicate Zr reinforced	184.73 ± 13.63	c	16.03	190.79
Celtra Duo HT	lithium Silicate Zr reinforced	174.15 ± 21.76	c	9.64	183.22
n!ce LT	lithium Aluminosilicate	172.12 ± 11.98	c	16.99	177.43
IPS Empress CAD HT	leucite-reinforced glass ceramic	131.16 ± 13.33	d	11.36	137.09
IPS Empress CAD LT	leucite-reinforced glass ceramic	127.65 ± 11.09	d	13.34	132.64

One-way ANOVA, Tukey test (*p* < 0.05). Different letters mark statistically significant differences. Legend: *σ* = Flexural strength (mean and standard deviation); *Sig* = Significance; *m* = Weibull modulus; *σ*_0_ = Weibull characteristic strength. The same letter of significance indicates no statistically significant differences.

**Table 3 materials-15-07834-t003:** Recommended clinical indications of ISO standards 6872:2015 (Classes 1, 2, and 3).

Class	Recommended Clinical Indications	Flexural Strength Minimum (Mean) MPa
1	(a) Ceramic for coverage of a metal framework or a ceramic substructure. (b) Monolithic ceramic for single-unit anterior prostheses, veneers, inlays, or onlays	50
2	(a) Monolithic ceramic for single-unit, anterior or posterior prostheses adhesively cemented. (b) Partially or full covered substructure ceramic for single-unit anterior or posterior prostheses adhesively cemented.	100
3	(a) Monolithic ceramic for single-unit anterior or posterior prostheses and three-unit prostheses not involving molar restoration adhesively or non-adhesively cemented (b) Partially or fully covered substructure for single-unit anterior or posterior prostheses and for three-unit prostheses not involving molar restoration adhesively or non-adhesively cemented	300

**Table 4 materials-15-07834-t004:** Results of tested materials ranked by translucency (CR).

		Translucency	
Material	Type	CR	*Sig*
Celtra Duo LT	lithium Silicate Zr reinforced FC	74.1 ± 1.1	a
Initial LiSi Block LT	lithium disilicate FC	74.0 ± 1.1	a,b
n!ce LT	lithium Aluminosilicate FC	73.3 ± 0.8	a,b
IPS Empress CAD LT	leucite-reinforced glass ceramic	73.0 ± 1.5	a,b
n!ce HT	lithium Aluminosilicate FC	72.4 ± 1.0	b,c
e.max CAD LT	lithium disilicate	71.3 ± 1.1	b,c
e.max CAD HT	lithium disilicate	65.2 ± 0.9	d,e
Initial LiSi Block HT	lithium disilicate FC	63.8 ± 1.1	d,e,f
IPS Empress CAD HT	leucite-reinforced glass ceramic	63.6 ± 1.2	e,f
Celtra Duo HT	lithium Silicate Zr reinforced FC	62.2 ± 0.8	f

One-way ANOVA, Tukey test (*p* < 0.05). Different letters mark statistically significant differences. Legend: CR = Contrast Ratio (mean and standard deviation); *Sig* = Significance. The same letter of significance indicates no statistically significant differences; FC = Factory Crystallized.

## Data Availability

The data presented in this study are available on request from the corresponding author. The data are not publicly available due to the university’s policy on access.

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
