# Peer review of "Factory Crystallized Silicates for Monolithic Metal-Free Restorations: A Flexural Strength and Translucency Comparison Test"

_materials, 2022, doi:10.3390/ma15217834_

Round 1

Reviewer 1 Report

This paper investigated the flexural properties and translucency of different lithium silicate-based CAD/CAM materials. Generally, the write-up of the paper is good but could get better after some corrections to the English language. Please see the comments below.   

Abstract:

-          The specimens’ dimensions are missing, as well as the statistical analyses used.

-          The abbreviation for IPS e.max CAD has been mistakenly reported as “LS” in the abstract.

Introduction:

-          Line 36, the proposition “towards” is incorrectly used.

-          The authors could add few sentences about the availability and differenced between the variable translucencies of the materials in the introduction section to justify the aim of the study and the 2nd hypothesis.

-          The acronym HT and LT have been mentioned initially in introduction section without stating the full name.  

Materials and methods:

-          This section is nicely stated and detailed. Kindly provide the sample calculation method.

-          For n!ce material, the composition does not include fluoride. Conversely, it has high percentage of alumina, so how is it lithium fluorosilicate and not lithium aluminosilicate?

-          How were the tiles used for translucency (15*15*1 mm) cut from the C14 blocks?

-          What was the polishing standardization measures taken to ensure all specimens had the same surface finish?

Results:

-          In Table 2, the decimal separator “.” Has been replaced by a comma “,” in some cells.

-          Lines 202-204 in the result section is better moved to discussion section.

-          What do you mean by original structure in SEM sections? For n!ice SEM images, Lines 235-236, could the globular porous structures as you referred to be alumina particles?

Discussion:

-          Generally, the discussion includes adequate comparisons with previous studies. However, there is a slight lack of explanation of the study’s own results and possible causes for such significant/non-significant differences between tested materials.   

-          Lines 263-266, regarding the 3-PBT, can be moved to limitations of the study.

-          The author can refer to the following references that tested the pressed version of Li-Si, either in 3-PBT or biaxial flexural strength for comparison:

o    Al-Thobity AM, Alsalman A. Flexural properties of three lithium disilicate materials: An in vitro evaluation. Saudi Dent J. 2021 Nov;33(7):620-627. doi: 10.1016/j.sdentj.2020.07.004. Epub 2020 Aug 6. PMID: 34803310; PMCID: PMC8589621

o    Katsura Ohashi, Yuka Kameyama, Yuuki Wada, Tomoyasu Midono, Kaori Miyake, Karl-Heinz Kunzelmann and Tomotaro Nihei. 2017. Evaluation and comparison of the characteristics of three pressable lithium disilicate glass ceramic materials.”, International Journal of Development Research, 7, (11), 16711-16716.

-          Line 292, the value 206±27.7 MPa is for HT?

-          Line 321, what do you mean by “e.max CAD, the only fully crystallized material”? Also, the references mentioned [31,32] did not test e.max CAD.

-          Line 322, “Conversely to other materials whose findings are reported in literature [31,33]”. Are you referring here to the reported literature with regards to flexural properties or CR? If it is for CR, then reference [33], does not fit as it reports flexural properties.   

-          Is it possible to draw any correlations between the SEM findings and the results for both tested properties?

-         Include the limitations of the study.

-         The last paragraph of the discussion section can be combined with the conclusions section.

Conclusions:

-          The conclusions are written well and covered all aspects of results.

-          What does PC stand for?

-          Line 374-375, the sentence is missing the referral to the tested property. Flexural strength, translucency or both.

References:

-          References [1], [2], [8], [21] are not mentioned in the text.

-          References [9] and [31] are duplicates.

Author Response

Abstract:

-          The specimens’ dimensions are missing, as well as the statistical analyses used.

We are aware of these shortcomings, and we agree with the reviewer. Unfortunately, even if specimens’ dimension and statistical analysis used were present in a preliminary version of the abstract, the strong limitation of the abstract to 200 words forced us to cut some parts. Reporting specimen dimensions and statistical analysis in the abstract means necessarily to take off other information.

-          The abbreviation for IPS e.max CAD has been mistakenly reported as “LS” in the abstract.

You are right, sorry for the typo.

Introduction:

-          Line 36, the proposition “towards” is incorrectly used.

We changed into “in comparison to”.

-             The authors could add few sentences about the availability and differenced between the variable translucencies of the materials in the introduction section to justify the aim of the study and the 2nd hypothesis

Following the suggestions of the reviewer, one sentence was added and other slightly rephrased.

-          The acronym HT and LT have been mentioned initially in introduction section without stating the full name. 

By adding the sentence about the former point, we also solved the present question. Acronyms are now presented at their first mention stating the full name.

Materials and methods:

-          This section is nicely stated and detailed. Kindly provide the sample calculation method.

The sample size was determined with reference to previously published studies with similar methods and in consideration of the minimum number of specimens needed for the Weibull statistics according to ISO 6872:2015. As a further check of the adequacy of the sample size, it was noticed that in each One-Way ANOVA the power of the test was above 0.8 threshold with an alpha error of 0.05.

-          For n!ce material, the composition does not include fluoride. Conversely, it has high percentage of alumina, so how is it lithium fluorosilicate and not lithium aluminosilicate?

You are right. We used a definition retrieved somewhere (but surely coming from the Company), but we noticed that now their definition reports aluminosilicate. We have replaced it in the table. Thank you.

-          How were the tiles used for translucency (15*15*1 mm) cut from the C14 blocks?

The method for producing the tiles for translucency measurements has been better described in the text.

-          What was the polishing standardization measures taken to ensure all specimens had the same surface finish?

The polishing standardization was already reported, but on the basis of this request it has been further clarified.

Results:

-          In Table 2, the decimal separator “.” Has been replaced by a comma “,” in some cells.

Corrected, thank you.

-          Lines 202-204 in the result section is better moved to discussion section.

For unknown reason, there is slight shift between reviewer and authors manuscript line numbers, therefore we are not sure of what the reviewer refers to.

-          What do you mean by original structure in SEM sections? For n!ce SEM images, Lines 235-236, could the globular porous structures as you referred to be alumina particles.

For original we mean that it is different from what previously observed in other Lithium silicates glass ceramics (we did not find in literature any previous SEM observation of N!ce …). However, the sentence has been slightly changed adding a comment to clarify.

Concerning the globular particles, while for the LiSi we had additional information from the manufacturer, for n!ce we hadn’t, so we preferred to avoid speculations. They might be alumina particles as the reviewer writes, or, in a similarity with LiSi, they might be the results of the aggregation of Lithium disilicate crystals and glass matrix …

Discussion:

-          Generally, the discussion includes adequate comparisons with previous studies. However, there is a slight lack of explanation of the study’s own results and possible causes for such significant/non-significant differences between tested materials.  

Two paragraphs were added in this regard.

-          Lines 263-266, regarding the 3-PBT, can be moved to limitations of the study.

Limitations of the study have been reorganized and should now read clearer.

-          The author can refer to the following references that tested the pressed version of Li-Si, either in 3-PBT or biaxial flexural strength for comparison:

o    Al-Thobity AM, Alsalman A. Flexural properties of three lithium disilicate materials: An in vitro evaluation. Saudi Dent J. 2021 Nov;33(7):620-627. doi: 10.1016/j.sdentj.2020.07.004. Epub 2020 Aug 6. PMID: 34803310; PMCID: PMC8589621

o    Katsura Ohashi, Yuka Kameyama, Yuuki Wada, Tomoyasu Midono, Kaori Miyake, Karl-Heinz Kunzelmann and Tomotaro Nihei. 2017. Evaluation and comparison of the characteristics of three pressable lithium disilicate glass ceramic materials.”, International Journal of Development Research, 7, (11), 16711-16716.

We thank the reviewer for this indication. We will for sure read the papers. However, as it is known that CAD and Pressed materials have to be considered different materials, e.g. [Fabian-Fonzar et al. A. Flexural resistance of heat-pressed and CAD-CAM lithium disilicate with different translucencies. Dental Mater 2017;33:63-70] we deliberately avoid in the discussion to refer to pressed materials.

-          Line 292, the value 206±27.7 MPa is for HT?

We removed this value as in the cited paper only HT was tested.

-          Line 321, what do you mean by “e.max CAD, the only fully crystallized material”? Also, the references mentioned [31,32] did not test e.max CAD.

There is a general confusion in the terminology for some of the materials. Some Companies refers material to be crystallized with thermal treatment as “Partially Crystallized”. But of course, if you test them after thermal treatment, thus after crystallization, you can’t call them “Partially Crystallized” as after the thermal treatment they are “fully crystallized”.

To try to overcome this confusion, the sentences was changed into “e.max CAD, the only material requiring thermal treatment”.

-          Line 322, “Conversely to other materials whose findings are reported in literature [31,33]”. Are you referring here to the reported literature with regards to flexural properties or CR? If it is for CR, then reference [33], does not fit as it reports flexural properties.  

References were renumbered (see answer to the previous and to the last question)

-          Is it possible to draw any correlations between the SEM findings and the results for both tested properties?

Reasonably yes for flexural, as it mainly depends on crystals number, dimension and size. Much more difficult for translucency, because to increase/decrease the translucency different strategies can be used (changing the type of filler, adding pigments, modifying the glass matrix…). In the absence of information from the manufacturers, any speculation is tentative and rough, and it should be at least supported by microanalysis …

However, based on the request of the reviewer, two paragraphs in this regard were added.

-         Include the limitations of the study.

Limitations of the study have been reorganized and should now read clearer.

-         The last paragraph of the discussion section can be combined with the conclusions section.

Unfortunately, we got different reports on this … almost impossible to satisfy all the reviewers as they asked for conflicting changes …

Conclusions:

-          The conclusions are written well and covered all aspects of results.

Thank you

-          What does PC stand for?

Again, there were problem with definitions (see answer to line 321). We initially defined this new category differently, but then we switched to Factory Crystallized. One of the acronyms remained … now it is changed to FC

-          Line 374-375, the sentence is missing the referral to the tested property. Flexural strength, translucency or both.

Flexural strength. We clarified it, thanks.

References:

-          References [1], [2], [8], [21] are not mentioned in the text.

-          References [9] and [31] are duplicates.

We apologise for these mistakes. Few papers were added at the very last moment before submission and the renumbering was not made accurately. Sorry for this. Now everything should be fixed.

The authors like to thank the reviewer for the complete and constructive review.

Reviewer 2 Report

Overall, the quality of the manuscript is satisfied. Nonetheless, here are few issues to be addressed by the authors:

1- self citation is more than the normal case in research activities. The first author, Vichi A. was found to cited eight (8) of his/her work while only 37 references were cited.

2- SEM results were presented. However, there is no any clear explanation related to the effect SEM with flexural strength. It is well known that microstructural features such as grain size, grain shape, porosity in the microstructure,  grain boundary migration, ... have a great influence on  the flexural strength. None of these parameters have been addressed throughout the manuscript.

These parameters in the microstructure greatly affect the flexural strength values.

3- Again, the possible effect of microstructure features was not reported on  the translucency of the tested materials.

4-Schematic representation for 3 point bending test is preferred and why it was selected over 4 point bending test (not just because of easier comparison with the literature as the authors stated)?? Taking into account the limitations as mentioned by authors when 3 PBT is used instead of the other two flexural strength testings.

Author Response

  • self citation is more than the normal case in research activities. The first author, Vichi A. was found to cited eight (8) of his/her work while only 37 references were cited.

As the reviewer states, self-citation is more than normal in research activity. Therefore, even if 7 papers of the first author are cited (actually seven, one was a duplicate, sorry for this), providing they are relevant to the text and published in IF journals we don’t see any reason to remove/replace them. They only indicate that the first author worked and published extensively on the subject

2- SEM results were presented. However, there is no any clear explanation related to the effect SEM with flexural strength. It is well known that microstructural features such as grain size, grain shape, porosity in the microstructure,  grain boundary migration, ... have a great influence on  the flexural strength. None of these parameters have been addressed throughout the manuscript. These parameters in the microstructure greatly affect the flexural strength values.

3- Again, the possible effect of microstructure features was not reported on the translucency of the tested materials.

Based on the two requests of the reviewer, in order to improve this aspect two paragraph were added to the text.

4-Schematic representation for 3 point bending test is preferred and why it was selected over 4 point bending test (not just because of easier comparison with the literature as the authors stated)?? Taking into account the limitations as mentioned by authors when 3 PBT is used instead of the other two flexural strength testings.

A schematic representation of 3PBT was added as requested. Concerning the selection of the strength test, we had to select one of the three conventionally performed (3-PB, 4-PB, Biaxial). All of the three methods have advantages and disadvantages. In this regard, we did not report in the text “because of easier comparison with the literature”, rather than we selected 3PBT it as it is i) the most commonly performed, ii) highly standardized, iii) has an easier setup iv) allows a higher possibility of comparison with other studies. However, a further comment on the limitations of 3-PBT has been added.

Reviewer 3 Report

Dear Authors,

I appreciate the efforts involved in the study, however I have some reservations prior to acceptance of your manuscript.

Author Response

Dear Authors,

I appreciate the efforts involved in the study, however I have some reservations prior to acceptance of your manuscript.

Dear Authors,

The study aimed at measuring some mechanical and optical properties for three different factory crystallized lithium silicates. The authors have executed and described the methodology in a clear manner.

General comments:

  1. The qualitative examination of specimens under SEM do not add any strength to the obtained results; the micro investigation is incomplete and does not support the mechanical and optical properties measured.

On the basis of this comment and of other reviewers, we added some comments on SEM observation.

  1. The results of Weibull modulus have not been substantiated anywhere in the discussion section. Higher the value, more uniform is the distribution of defects which describes the lifetime and frequency failure of brittle materials.

A paragraph on Weibull outcome has been added

Specific comments:

  1. Abstract:
    a. Page 1 - Line 20 - The abbreviation “3-PBT” must be expanded
    b. Results – Instead of mentioning the values in descending order, the significantly different groups could have been highlighted with statistical tests mentioned in

We are aware of some limits of the abstract (e.g. specimen dimensions, statistical analysis used, results not easy to read, excessive use of acronyms) but we kindly ask the reviewer to consider that there is a 200 words limit that is very narrow and something has to be sacrificed.

  1. Introduction:
    a. Page 1 - Para 1 – The sentences should be cited; the inline citations 1 and 2 are missing
    b. Page 2 - Para 1 – The inline citation 8 is missing

Few references were added just before submission and not all the citations were properly renumbered. We apologize for this. We have now rechecked all the citations and they should be correct.

         c. Page 2 - Para 2 – The rationale for the study should be clearly mentioned prior to the aim of study.

The aim of the study has been rephrased.

  1. Results
    a. Table 2 –
    i. The values of Weibull statistics is not expressed uniformly; the period and comma were used for values.

We are sorry for these errors in formatting. We rechecked the table and should now be correct.

         ii. The footnote contains σ0 that appears twice, needs to be verified.

We rechecked the footnote, but it looks that there is a σ and a σ0 …

         iii. The footnote should include the significant p value

The p value was added to the text and additional information were placed in the footnote

         b. Table 4 –

         i. The footnote should include the significant p value

The p value was added to the text and additional information were placed in the footnote

         ii. The footnote should expand the abbreviation PC and FC

Only FC was left (see comment to conclusions). The abbreviation has been described in the footnote.

          c. Page 5 – Para 2 – line 192 – Table 3 could have been cited along this sentence instead of doing it later in paragraph no. 3

Generally speaking, we tried to place tables in a way that they are not split in different pages. At present, due to the several changes made, that have to be tracked, it is difficult to visualize the final aspect of the paper. We will for sure keep this indication of the reviewer for the final version, cleaned from track changes.

  1. Discussion
    a. The SEM finding were not discussed

Some sentences were added to discuss SEM findings.

         b. Page 8 – Para 2 – The inline citation 21 is missing

Once again, we apologize for the wrong renumbering, that was addressed.

         c. Page 10 – Para 1 – line 343 – The sentence stating with “A limitation….” – It needs to be revised   grammatically.

On the basis of the request of this and other reviewers, the part concerning limitations was expanded and completely rewritten.

          d. Page 10 – Para 3 & 4 – Both paragraphs appear to be authors’ objective opinion and not supported by evidences.

This comment of the reviewer it is not clear. Anyway, page 10 contains the definition of translucency and some comments on the measured differences between HT and LT formulations. The comments are based on the literature concerning perception of the differences in translucency, that is cited. Unfortunately, only paper could be identified that reported the human eye perception of differences in translucency based on CR. Therefore, the comments are supported by literature.

  1. Conclusion
    a. The abbreviation PC is not mentioned anywhere in the text other than table 4.

This is right. There is a general confusion in the terminology for some of the materials. Some Companies refers material to be crystallized with thermal treatment as “Partially Crystallized”. But of course, if you test them after thermal treatment, thus after crystallization, you can’t call them “Partially Crystallized” as after the thermal treatment they are “fully crystallized”. To try to overcome this confusion, the definition of Partially Crystallized (PC) was removed throughout the text, a definition of FC (Factory Crystallized) has been added, and the sentences was changed into “e.max CAD, the only material requiring thermal treatment”.

          b. The conclusion could be made short and directly addressing the aim of study

Unfortunately, we had different comments by the reviewers on this aspect. Almost impossible to address all the requests as they are conflicting. Sorry for this.

          c. Reference – All references should be verified for accuracy.

All the references were rechecked.

Reviewer 4 Report

This is an interesting study but needs significant corrections and editing of the manuscript.

Abstract

The abstract needs significant improvement.

AT someplace”,” is there and someplace”.” is there at the numbering. So, the authors should follow the journal’s guidelines.

Add objectives of this research.

Introduction

In the first paragraph of the manuscript, it is better to add the relevant latest references.

https://pubmed.ncbi.nlm.nih.gov/34500741/

Materials and Methods

Please provide about sample size.

Did the researchers do the experiments according to the ISO standard testing? As they mentioned in the conclusion.

Please add references for the fractural strength and Weibull characteristic strength equation.

Results

Figure 1 should be edited to make it better fitting in the manuscript.

Discussion

The Discussion needs to be elaboration comparing with proper studies.

Please discuss the marginal fit which has an impact on the esthetics.

In Vitro Microscopic Evaluation of Metal-And Zirconium-Oxide-Based Crowns’ Marginal Fit. https://www.scielo.br/j/pboci/a/zJ9XM9zHNHx773pKMbdgFsS/?format=html&lang=en

Does the sintering protocol affect color stability? Please discuss.

Please add the limitations of this research.

Conclusion

It is long. Present the conclusion in a better way and make it concise.

Overall

Major revision with English corrections is needed.

Re-access is needed after the revision.

Author Response

Abstract

The abstract needs significant improvement.

AT someplace”,” is there and someplace”.” is there at the numbering. So, the authors should follow the journal’s guidelines.

We made the requested changes. Thank you.

Add objectives of this research.

We are aware that the abstract is missing some parts like a better description of objectives, statistical test used, dimension of specimens and other. Unfortunately, the 200 words limit is very narrow, and we had necessarily to shorten or cut some parts.

Introduction

In the first paragraph of the manuscript, it is better to add the relevant latest references.

https://pubmed.ncbi.nlm.nih.gov/34500741/

The reference indicated is concerning with zirconia, while the present paper is about lithium disilicates.

Materials and Methods

Please provide about sample size.

The sample size was determined with reference to previously published studies with similar methods and in consideration of the minimum number of specimens needed for the Weibull statistics according to ISO 6872:2015. As a further check of the adequacy of the sample size, it was noticed that in each One-Way ANOVA the power of the test was above 0.8 threshold with an alpha error of 0.05.

Did the researchers do the experiments according to the ISO standard testing? As they mentioned in the conclusion.

As reported in the text, the experimental part for flexural test has been performed following ISO standard 6872:2015.

No ISO standards are available for Translucency test.

Please add references for the fractural strength and Weibull characteristic strength equation.

References have been added.

Results

Figure 1 should be edited to make it better fitting in the manuscript.

We followed strictly the template, trying to show the images as much large as possible with two images (HT and LT) in a row for each of the 5 materials, for comparison purposes. If the reviewer could be more clear on what he/she means for “better fitting” we can try to improve it.

Discussion

The Discussion needs to be elaboration comparing with proper studies.

Please discuss the marginal fit which has an impact on the esthetics.

In Vitro Microscopic Evaluation of Metal-And Zirconium-Oxide-Based Crowns’ Marginal Fit. https://www.scielo.br/j/pboci/a/zJ9XM9zHNHx773pKMbdgFsS/?format=html&lang=en

The suggested paper is about Zirconia that is not the subject of the present paper. Moreover, the present paper is not about “marginal fit” and it is not about “esthetics”, rather about flexural strength and translucency.

Does the sintering protocol affect color stability? Please discuss.

This paper is not about Zirconia, rather Lithium (di)silicates, thus there is no sintering protocol. Moreover, in this paper color stability has not been tested, rather flexural strength and translucency.

Please add the limitations of this research.

Limitation of the research have been reorganized and should now read clearer.

Conclusion

It is long. Present the conclusion in a better way and make it concise.

Unfortunately, we had conflicting indications from the reviewers on this aspect.

Round 2

Reviewer 2 Report

Authors' corrections are acceptable

Reviewer 4 Report

It is better to add on the mechanical properties of the recent translucent zirconia materials.

https://doi.org/10.3390/molecules26175308